# The Significance of Angling in Stress Reduction during the COVID-19 Pandemic—Environmental and Socio-Economic Implications

**DOI:** 10.3390/ijerph19074346

**Published:** 2022-04-05

**Authors:** Emil Andrzej Karpiński, Andrzej Robert Skrzypczak

**Affiliations:** Department of Tourism, Recreation & Ecology, Institute of Engineering and Environmental Protection, University of Warmia and Mazury, Oczapowskiego St. 5, 10-719 Olsztyn, Poland; sandacz@uwm.edu.pl

**Keywords:** recreational fishing, aquatic environment, health, COVID-19, stress reduction, socio-economic conditions

## Abstract

Recreational fishing and other outdoor recreational activities have been proven to have positive effects on mental health, including neutralizing pandemic stress. This study aims to identify the perceptions and behavior of recreational anglers during the COVID-19 pandemic along with identifying the factors that determine attitudes. This study is essential for understanding the complex economic, social, and environmental implications associated with a pandemic. Perceptions of pandemic stress reduction were confirmed by 63.8% of anglers, and nearly 98% felt there was no risk of infection from fishing. These feelings were most strongly positively associated and explained by a preference for fishing with friends and family and the frequency of fishing. Over 26% of respondents fished more frequently during the pandemic. Additional free time and the need to escape the pandemic media hype were the main factors for the increase in angler activity. The balance of benefits from changes in angler pandemic behavior was ambiguous. This was determined by considering the potential increase in pressure on the environmental resources that anglers use. Given the positive effects of angling on stress reduction, it is not advisable for policymakers to restrict recreational fishing access. Instead, best management practices should reduce sanitary bottlenecks to increase safety on fishing grounds.

## 1. Introduction

### 1.1. Social and Environmental Impacts of the Pandemic

The COVID-19 pandemic hit the world in early 2020 and resulted in a great extent of negative implications in both economics, e.g., [1,2], and society, e.g., [3]. The pandemic was very complex and had an ambiguous environmental impact, e.g., [4,5]. The pandemic made people realize the value of health protection. Although lockdown policies seemingly promoted health maintenance, they also had many side effects. Among the negative health effects, the most common were alcohol overuse, e.g., [6,7], excessive weight gain e.g., [8,9,10], domestic violence e.g., [11,12], and other severe psychological effects [13]. 

One way of neutralizing the direct and indirect social effects of the pandemic was participating in recreation and mass sports. Due to the implementation of sanitary and public health regime rules in most countries of the world, leisure and sports activities were partially or temporarily banned—especially indoors. These lockdown policies caused recreational activities to decline in most cases, e.g., [14]; however, long-term home isolation and reduced socializing led many to participate in recreational activities, especially outdoors, once restrictions were loosened [15,16]. Open-air outdoor recreation has been proven to have positive effects on mental health and can reduce stress, e.g., [17,18], including pandemic-related stress, e.g., [19]. 

### 1.2. Fishing during a Pandemic—A Literature Review

Angling is one of the most popular outdoor recreational activities [20]. The research so far shows that it is regarded as “giving a sense of security” in the pandemic. This is the result of anglers’ subjective perspectives, and references to objective reasons are related to best angling practices and knowledge of the aerosol or fomite transmission of viruses, such as SARS-CoV-2 [21,22,23]. This is because of two key factors. The first is an open-air environment, which, according to research [22,24], significantly reduces the chance of a COVID-19 transmission. The second is its very specificity, which almost naturally enforces social distance, already referred to in the literature as “social fishstancing”, e.g., [21]. The legal solutions in some regions, or of specific fisheries, often enforce a minimum distance between active anglers, and this legislation was in force before the pandemic [25,26]. The competition regulations on this issue also allowed distance to be maintained or were changed to allow this to occur [27,28]. The above specificity has led to an increased interest in fishing participation among many people. Anglers’ choices, for example, those due to reduced mobility, have also changed due to the pandemic [21,23,29,30]. Nevertheless, many concerns are associated with the spread of COVID-19 in recreation, even outdoors [31], and it is essential to always take these risks into account. Moreover, despite the relatively mild way in which the pandemic has affected angling as a whole, anglers have not been spared from its risks and restrictions. For example, sharing a fishing spot, transportation, or boat began to become riskier [32]. Due to lockdown, there was also a lack of access to hotel services [33] or new gear and baits through disruptions of supply chains [34,35].

### 1.3. Research Objectives

This study aimed to identify anglers’ activities, including their perceptions and behavior during the COVID-19 pandemic and the factors that caused these attitudes. Understanding that angling is based on a complex system of motivational needs (e.g., the leading need is to escape to nature [36,37]), it was assumed this form of recreation may reduce pandemic stress. It was also hypothesized that this sense of stress reduction was not dependent on socio-demographic factors or preferences and involvement in fishing. To verify these hypotheses, we asked the following: (1) to what extent do socioeconomic factors, as well as preference and engagement factors, explain the variability in anglers’ feelings and behavior during the pandemic and (2) what factors determine the attitudes of anglers who feel most strongly their hobby has a positive impact on reducing pandemic stress? Understanding this is necessary for the rational management of water resources and their surroundings, which have proven to be, on the one hand, a relatively safe environment for recreation and, on the other hand, a critically needed resource for the psycho-physical regeneration of pandemic-affected populations. Changes in the behavioral patterns of society due to pandemics, in general, as well as policies and recommendations triggered by the pandemic, are also important in this context. This study is part of an important trend in understanding the complex economic, social, and environmental implications caused by the spread of the SARS-CoV-2.

## 2. Materials and Methods

### 2.1. Design of the Questionnaire and Data Collection

We used WAIs (Web-Assisted Interviews), which were constructed on the English and Polish Google Forms platforms (https://docs.google.com; last access on 1 September 2021). WAI surveys are easy, cheap, very efficient, and less error-prone compared to traditional questionnaires. They allow for faster access to and analyses of data. The weaknesses of WAIs are that we never know who is filling in the survey (which is always the case with anonymous surveys), and they are accessible only to people with access to the Internet. However, this drawback is of marginal importance in this study because it was within the methodological assumptions [38].

Our survey was based on 24 questions. The first 11 measured socio-demographic (e.g., age and education), economic (e.g., income and angling expenses), and engagement in angling (e.g., association, avidity, and experience) characteristics. In these questions, respondents were asked to indicate the number closest to the truth according to their best knowledge or to use a yes/no option. The exemptions from this rule were questions about place of residence and education status. In the place of residence question, there was a “village” option in addition to numeric ranged values. In the education question, there were 3 options to choose from the following: primary, secondary, and tertiary, which were then converted into years of education (numerical data) before being included in the data analysis. The second part obtained opinions on attitudes and behavior regarding living and fishing during the SARS-CoV-2 pandemic (15 items). Answers to these questions were ranked on a typical 5-point Likert scale, where “1” was the equivalent of “I strongly disagree”, “5” meant “I strongly agree”, and “3” meant “I have no opinion, or it is difficult to determine” (neutral opinion). This scale is widely used in surveys designed to measure respondents’ opinions [39,40]. 

The questionnaire was disseminated via various popular websites and social media platforms (e.g., angling associations and clubs, Facebook groups and fan pages, anglers’ discussion groups, and internet forums) in English and Polish. Survey respondents were encouraged to distribute the survey among their angling communities with the restriction of completing the survey only once. This approach was intended to create viral content, which is a modern and effective way to reach the largest number of people [41]. The survey was widespread from July to August 2021. The assumption was that surveying at a time of relatively low levels of infection (summer) was expected to result in a lack of impact of lockdown-like policies on the respondents’ judgment. This effect was further amplified by surveying after a year of operating in a so-called “pandemic environment.” The questionnaire was fully voluntary, anonymous, and not limited by age. The time required to complete the survey was about 5 min. The full questionnaire is available in Appendix A.

In total, 564 respondents provided complete answers to the survey questions. This number of completed questionnaires was large enough to apply to the population as a whole. The margin of sampling error (MoE) at confidence-level α = 95% was calculated (Appendix A). It was ±4.13% for the whole sample and ±3.6% to ±8.0% for each socio-demographic subgroup. A rule of thumb researchers considers adequate to reflect the opinion of the entire population is a MoE of 4–8% [42,43]. Given the type of survey distributed, the sample included people who were active on angling social media. On this basis, it should be assumed that the survey mainly covered Europeans residing in EU countries and the UK. Yet, determining the response rate seems impossible due to, on the one hand, a desire to create viral content and, on the other hand, the ephemerality of this sort of data. Cautious estimates based on the reaches of the groups, the numbers of members, and their activity suggested that the number of anglers was between 10,000 and 500,000.

### 2.2. Statistical Analyses

Socio-economic, demographic, and engagement data were examined using frequency table characteristics. The percentages of respondents with statements related to angler preferences and behavior, expressed on a Likert scale, were calculated similarly; we assumed a non-linear distribution of the Likert scale [44]. Non-parametric tests are better for analyzing ordinal scale data, such as Likert data [45].

We examined the differences between the preferences and behavior of the anglers who indicated stress reduction and those who did not by using a non-parametric Kruskal–Wallis ANOVA rank-sum test for independent samples (*p* < 0.05). However, due to the large sample size, the variation in respondents’ opinions was chosen to be shown not by median but by mean and standard deviation. All statistical significance tests were performed using Statistica version 13.3 software. 

Redundancy analysis (RDA) was used to identify relationships between anglers’ feelings and behavior during the pandemic and socio-economic and demographic factors, as well as including their preferences and engagement in angling. RDA is a canonical form of principal component analysis and is one of the so-called linear techniques that can be used in socio-economic research [46]. This linear ordering method is appropriate when the greatest gradient in the dataset is less than 3.0 standard deviations [38]. The RDA space was used to explain the intensity of feelings of the following: reduction of pandemic stress by fishing; improvement in psychological condition and physical condition; fear of infection and getting sick; and acceptance of COVID-19 vaccination. In contrast, behaviors were explained by the following: increased frequency of fishing during the pandemic and reduced contact with family and friends. Anglers’ responses were compositional and had a gradient of 0.2 SD unit lengths, so a linear method better explained the data. 

Each variable that explained anglers’ feelings and behavior was tested for statistical significance using Monte Carlo tests (499 random permutations). Data were normalized using the log (x + 1) transformation [47]. All variables explained a significant amount of variation and were statistically significant (*p* < 0.05). The explanatory variables (socio-demographic factors, preferences, and engagement indicators) were selected based on a variance inflation factor (VIF) of less than 10. During the RDA analysis, the numbers of response data (reactions and behavior) and explanatory variables were verified each time based on the values of the correlation coefficients of the explanatory variables and VIF. The purpose of this verification was to obtain the maximum value of the percentage of the explained total variance of response data [47]. Finally, 10 explanatory variables were implemented into the ordinal space, including age, education, income, domicile, costs (angling expenses), distance (to angling site), angling experience (angling seniority), avidity (angling frequency), and preference for fishing alone or with family or friends. RDA was performed using the Canoco version 5.11 software.

## 3. Results

### 3.1. Summary of Responses—Anglers across Sociodemographic, Economic, and Engagement Factors

Most of anglers surveyed were in the definitive working-age group (25–65 years) representing 79.3% of the surveyed population (Appendix A). The age group potentially most vulnerable to the effects of a viral infection (older than 65 years) comprised 6.4% of the anglers. The most common income level (51.6%) was between 6 and 12 thousand Euros per year, and about one-third (34.5%) lived in small settlements (villages and towns up to 5000 people); 27.7% lived in large cities (over 100,000 people). Most of the respondents had a secondary (44.1%) or higher (39.4%) educational level. Their fishing experience ranged from novice to over 40 years of fishing, with the largest number of anglers (49%) having between 10 and 30 years.

The anglers were fairly active with 55.4% of the anglers taking at least one fishing trip per week; 10% were occasional anglers (less than once a month). Seventeen percent claimed they spent less than EUR 100 per year on their hobby (equipment expenses, licenses, travel, etc.) while 13.3% spent more than EUR 1000. Most (56.9%) spent between EUR 100 and 500. Twenty-seven percent traveled less than 5 km to their favorite fishing spot, and 12.3% traveled more than 50 km. About 26.1% had fished within urban areas. The vast majority (81.4%) of the surveyed anglers were associated with clubs or organizations. Almost half of them considered themselves fish tourists (47.9%). The margin of sampling error (MoE) indicated that all socio-demographic subgroups were useful in reflecting the whole population. 

### 3.2. Anglers’ Life Attitudes and Behavior during the Pandemic

The anglers’ responded positively to stress-reducing capabilities of their hobby, with 63.8% agreeing (Table 1). They responded positively (61.1%) toward vaccination against COVID-19 with a mean of 3.56 ± 1.34. Simultaneously, the surveyed anglers were not very concerned about getting sick (or getting sick again); 16.5% of anglers were afraid of getting sick (or getting sick again) in general in their lives (mean 2.07) and only 2.1% of the respondents were afraid of getting sick while fishing.

More than a quarter (26.1%) claimed to fish more often during the pandemic. Only 7.5% and 13.9% reported improved mental and physical fitness, respectively. Some respondents indicated they limited their contact with family (40.5%) and friends (48%) during the pandemic (Likert-scale answers four and five), even though they indicated fishing with family (28%) and friends (47.9%). Forty-one percent declared they fished alone. Only 17% claimed that they had limited their outdoor activities in general (a Likert-scale mean score of 2.14 with a SD of 1.30). 

The anglers’ preferences and engagement characteristics as explanatory factors for their perceptions and behavior during the pandemic were illustrated in the ordination space (Figure 1). Table 2 shows the summary of the results of the RDA, including the eigenvalues, correlations, and percentages of variation explained by all the canonical axes. The correlation of all the axes was significant in the Monte Carlo permutation test (F = 6.8, *p* = 0.001). The sum of all the canonical eigenvalues was 0.8122. The two RDA factors explained a total of 73.23% of the variability of response data, with the RDA1 axis accounting for 59.27% of the total variance (in anglers’ life attitudes and behavior during the pandemic). The following were most strongly correlated with the RDA1 axis: increased frequency of angling (r = −0.836), acceptance of vaccination (r = 0.837), feeling reduced pandemic stress by angling (r = −0.746), fear of COVID-19 (r = 0.611), limiting contact with family (r = 0.729), and limiting contact with friends (r = 0.648). In contrast, feelings of an improved mental condition showed stronger correlations with the RDA2 axis (r = 0.603) than with the RDA1 axis (r = −0.541). The cumulative percentage variance of the fitted response data for both RDA axes was 90.16% (including RDA1 (72.97%)). Among the variables explaining the anglers’ perceptions and behavior during the pandemic, the strongest correlations with the RDA1 axis showed a preference for fishing alone (r = 0.7145) or with family (r = −0.7135) or friends (r = −0.8367), as well as angling seniority (r = 0.7177) and avidity (r = −0.5425). Thus, feelings of pandemic stress reduction were most strongly positively related to and explained by a preference for fishing with friends and family and an activity index, i.e., angling frequency. 

Furthermore, a preference for such behavior was positively related to the well-being of the improved mental states and physical conditions of the anglers. In contrast, feelings of improved mental and physical conditions were at odds with reduced contact with family and friends. Feelings of pandemic stress reduction were most strongly negatively related and explained by a preference for fishing alone. Simultaneously, anglers with this preference showed greater tendencies to reduce social contact. In contrast, the tendency to reduce contact with family and friends was not exhibited by anglers undertaking an increased fishing activity during the pandemic. This activity was negatively related to a fear of infection and was less frequent in the anglers with more seniority. The limiting factor for undertaking more activity was a greater distance from a favorite fishing spot. 

### 3.3. Factors of Perceived Pandemic Stress Reduction 

Respondents indicating stress reduction showed a significantly higher reduction of overall outdoor activity in their social space; however, they did fish more often (Table 3). Though they showed more of a fear of getting sick and less of an improvement in their physical and mental states, as well as a greater reduction in their contact with friends, these four indicators did not significantly differ from anglers reporting no reduction in pandemic stress. In contrast, the “no-reduction” group of respondents had statistically different and more positive attitudes toward vaccination and limited contact with their families to a lower extent.

Among the respondents who most strongly felt the effects of fishing on stress reduction, significant linear correlations were found between behavior toward reducing contact with family and friends (r = 0.924, *p* < 0.001), indicating a significant tendency to reduce social contact. Therefore, the answers to two questions were combined in the ordination space; “I have limited contact with my family during the pandemic period” and “I have limited contact with my friends during the pandemic period” were combined into “I have limited contact with my friends and family during the pandemic period.” Additionally, the feeling of an improved physical condition in this group of respondents was strongly correlated with the feeling of an improved psychological condition (r = 0.813, *p* < 0.001) and justified the desirability of looking for explanatory factors for the feeling of an overall improved psycho-physical condition. Therefore, in the ordination space, the answers to the questions “My mental condition has improved” and “My physical condition has improved” were combined into “My mental and physical condition has improved.” Four sociodemographic variables, i.e., age, education, income, and domicile, explained the behavior and responses among the anglers who most strongly felt the effects of angling on reducing COVID-19 pandemic stress (Figure 2). The total variance was 8.7426, and the correlation of all the axes was significant in the Monte Carlo permutation test (F = 20.20; *p* = 0.002). The sum of all the canonical eigenvalues was 0.8349 (Table 4).

The first two components of the RDA explained 82.56% of the total variance in the response data, of which the first axis accounted for 76.87%. More frequent angling during the pandemic (r = 0.76) and positive attitudes toward vaccination (r = −0.83) were most strongly correlated with this axis. In contrast, a fear of getting sick, improved mental and physical conditions, and reduced social contact were most strongly correlated with RDA2 (correlation coefficients r = 0.72, r = −0.48, and r = 0.62, respectively). The anglers’ ages and incomes were the sociodemographic factors most strongly correlated with the first axis (correlation coefficients r = –0.88 and r = −0.46, respectively). This means that the acceptance of vaccination increased with age and income. Such preferences were mainly the characteristics of the anglers: affiliated, practicing angling tourism, with the longest period of engagement (over 40 years), average spending (from EUR 100 to 250), and fishing relatively frequently (several times a month). 

However, the anglers’ places of residence and education were most strongly correlated with RDA2 (correlation coefficients r = 0.47 and r = 0.45, respectively). A greater fear of getting sick was correlated with a more intense limitation of social contact (r = 0.73) and was felt most strongly by the anglers from larger urban centers. Their favorite angling sites were usually in a city. They were characterized by relatively significant angling experience (ranging from 31 to 40 years) and usually fished at a low activity level (a dozen or so times a year). A more intense reduction in contact with family and friends was correlated with increased education and was most characteristic among the anglers with the least expenditure on their hobby (up to EUR 100). 

The anglers with wide ranges of seniority (ranging from 5 to 30 years) and who fished at the highest frequency (a few times a week) declared feeling an improvement in psycho-physical condition during the pandemic period. This was also the group that spent the most on their hobby, i.e., more than EUR 500, but did not prefer angling tourism. A feeling of improvement in one’s psycho-physical condition negatively correlated with the sizes of the anglers’ domiciles and did not coincide with an increased frequency of angling. An increase in angling activity during the pandemic with no apparent improvement in mental and physical conditions was found mainly among unaffiliated anglers, those with the least angling seniority (less than 5 years), and those who were angling the least frequently, usually a few times a year. Their attitudes were simultaneously negatively correlated with vaccination acceptance (r = −0.502) and limitation of social contact (r = −0.610). More frequent angling during the pandemic was most uncharacteristic of the oldest anglers (age 65+) with a correlation coefficient magnitude of r = −0.732.

## 4. Discussion

### 4.1. Angling as a Safe Activity during the Pandemic

Coping with stress and mental health problems brought on by limited social contact and isolation are important issues during the COVID-19 pandemic [13]. Research has indicated the positive stress reduction potential of outdoor recreational activities [17,18]. Additionally, recreation has a positive effect on neutralizing other pandemic effects, such as weight gain, e.g., [8,9,10]. As can be seen from this and other studies, e.g., [21], angling can be viewed as an activity that can significantly reduce the negative social and psychological impacts of pandemics.

One of the most important issues in this matter is the perception of angling as a safe form of recreation during a pandemic. Recent studies, e.g., [21,23], as well as this paper, have indicated that anglers perceive their hobby as a safe activity. Almost 98% were not afraid of getting sick during fishing, while two-thirds (66.5%) were not afraid of getting sick at all. This means that about 32% of the surveyed people who were in some way concerned about the possibility of contracting SARS-CoV-2 did not notice this concern while fishing. This perception can, of course, be due to many factors, both subjective and objective. We focused on the objective causes resulting from the characteristics of angling as a recreational activity.

There are two primary reasons why anglers may believe that angling will be a relatively “safe haven” during a pandemic. The first is the aspect of maintaining social distance. In most parts of the world, due to imposed restrictions during the COVID-19 pandemic, a safe distance between people (usually 1.5–2 m) was recommended. In fishing, this term was called social fishstancing. However, such, and even greater, distances were usually maintained for strictly practical purposes and were in force before the pandemic for pragmatic, not health, reasons. For example, in specific fisheries, e.g., [25], sporting rules during competitions regulating the size of a fishing spot, e.g., [27], or best practices required a distance between anglers not fishing together (e.g., in Poland, 10 m from the angler and 50 m from the boat) [26]. In addition to maintaining sanitary safety, fishstancing allows you to maintain freedom of movement and reduces the danger of accidentally hooking a fellow angler. One way to maintain distance during pandemics was to keep a “one-rod distance”, which was a popular method of promoting fishstancing on social media. Of course, it depends on the methods, techniques, and equipment used. There are situations where, due to the characteristics of the technique (e.g., passive carp fishing techniques), this rule is marginal. Other methods, e.g., shared fishing from boats, especially small ones, make it very difficult to keep your distance, but in that matter, some regulations were changed [28]. Worldwide policies and restrictions on being outdoors were more related to limiting mobility and avoiding crowds than to not being outdoors per se [48]; therefore, sports events and larger gatherings were prohibited. Anglers and other participants in wildlife activities (e.g., hunters and birdwatchers) changed their ways of conducting recreation to make it more local, closer to home, and in smaller groups, ergo without the crowds [21,49,50]. 

This is the second important factor in the relative safety of angling—the outdoors. Outdoor activities comprise most, if not all, angling activities in the world. According to research [21], the virus’s transmission in open areas is less likely to occur. There are positive conditions for not accelerating virus transmission through angling. Droplets mainly affect other organisms from distances of up to 1 m before dropping on the ground [51]. Thus, keeping up about 2 m of distance in most cases should be enough. 

The current factor that reduces pandemic fear is the ability to receive a vaccination [52]. An interesting, previously unexplored topic was the attitude of recreationists toward COVID-19 vaccination. Anglers who participated in our study, despite positive subjective feelings about the safety of their hobby, did not differ from the European average in their acceptance of vaccination for COVID-19. Nearly 60% had an unequivocally positive attitude toward it, while the European average was about 58% (full vaccination in December 2021). It is worth noting that at the time of the dissemination of the survey, the vaccination rates in Europe were about 30% [53]. There was a clear increase in the acceptance of vaccination with age, which seems very valuable in considering a current scientific consensus claiming that the elderly are most at risk of death from COVID-19, e.g., [54,55]. It was also noticeable that people with a higher acceptance of vaccination, for example, the group without a feeling of stress reduction through angling, at the same time were less likely to limit contact with their families. In the context of the current medical consensus, e.g., [56,57], which indicates that the vaccine does not prevent virus transmission, this seems quite unwise. This also demonstrates that our hypothesis of reducing pandemic fear through vaccination is plausible. An open question is whether such a sense of security brings more positives (mental health) or negatives (physical health) to a person. There was also a considerably positive response toward vaccination from anglers angling alone and those who needed to travel to greater distances to reach their favorite fishing spot. Additionally, acceptance also increased as earnings increased. In the group experiencing a reduction in stress, both affiliated anglers and angling tourism enthusiasts were more numerous in accepting vaccination than non-affiliated and non-tourist anglers. It is reasonable to assume that the acceptance of vaccination and the desire to return to a pre-pandemic lifestyle by undergoing it may stem from a desire to move freely in relative safety.

Another aspect worth mentioning is not participating in outdoor recreation with infections, which should be good practice for every form of recreation with or without a pandemic. However, a scientific consensus has identified a so-called “symptomless infection” with which we can transmit SARS-CoV-2 without having any symptoms. Anglers should behave responsibly by not going fishing when they feel that the risk of contracting an infection is very high in the days before fishing, even when there are no symptoms. Another issue is the use of the same accessories by two or more anglers. This is rather rare, so transfer by fomites is also limited.

### 4.2. Anglers’ Social Needs during a Pandemic

Reducing stress and improving physical and mental well-being by fishing were positively correlated with avidity and fishing with family and/or friends. Moreover, it was negatively correlated with fishing alone. Those most fearful of infection, often the most experienced (the elderly), isolated themselves more and limited contact with others and thus were less characteristic in the group that claimed to fish more often. Based on these results, it is reasonable to believe that individuals who isolated themselves in their daily lives as a result of lockdown policies sought compensation for their losses in social connectedness during recreational activities. 

Fishing is not a particularly popular recreational activity for fulfilling social needs. When fishing, anglers are usually more focused on their activities and observing nature than on other anglers. According to Karpiński and Skrzypczak, variations in this behavior may be related to angler specialization [58]. Our research indicates that when anglers are out fishing, they are most likely to go there alone. If they are not alone, they go with friends or family with whom, regardless of daily pandemic conditions, they socialize anyway. Simultaneously, they tend to limit meetings with family and friends (40% and 47%, respectively). This may suggest that the people with whom they fish are not included in this group. According to Skrzypczak and Karpinski, social connectedness is a differentiated need across socio-demographic factors, such as gender, income, education, activity, spending, experience, and membership in angling organizations [58]. Thus, while social ties are not a particularly important motivational factor, they probably take on a somewhat different meaning during a pandemic. Although most people limited gatherings, it has been noticed that among anglers, not limiting all possible contact had a positive effect on reducing stress. It is reasonable to assume that contact with family and friends was not reduced by the pandemic as much as the reduction of attending gatherings (with strangers). This, of course, did not eliminate risk, but it did reduce it by limiting it to a pool of people they knew well. Recreationists themselves consider gatherings in close communities safe [59].

Midway et al. concluded that the pandemic caused changes in the system of needs motivating angling. Their study found that the need for “contact with nature” was the only motivating factor in which no changes in preferences were noted [21]. The constancy and priority of this need, while the dynamics of the other motivations are variable, have also been demonstrated in other studies [36,37]. Therefore, pandemic changes in lifestyle, including lockdown with its consequences, should not be considered the main direct reasons for first contact with angling. They should be considered primarily as factors of increased activity by anglers who discovered a passion for fishing earlier. Such a conclusion is justified given the identified strong correlation of age with angling experience in all the studied age groups of anglers found in studies of their motivational needs [37].

### 4.3. The Effect of the Pandemic on Angler Behavior and Activity

If one way to reduce pandemic stress is through outdoor recreation, and fishing is considered a relatively safe form of recreation, this raises a question: could the search for safe forms of outdoor recreation contribute to the increased popularity of fishing worldwide? The occurrence of such a trend during a pandemic may result in an increased pressure on environmental resources. It is therefore important to identify the main factor of such a trend. 

Our survey showed that anglers have not decreased their overall outdoor activity significantly during the pandemic. To be precise, the group with a revealed stress reduction factor angled with approximately the same frequency. About 26.1% of all the surveyed anglers claimed to have increased frequency. A large proportion of respondents (37.8%) could not say whether the frequency of their fishing had increased or not. It is, therefore, reasonable to believe that their activity had remained at a similar level. In the group of anglers showing an increased frequency of angling, the least numerous anglers were those who had a long distance to their favorite fishing spot, were the most experienced, were the oldest (a high natural correlation between age and experience), and tended to fish alone.

As recent studies showed, closures did not necessarily result in a reduced angling pressure. In fact, many researchers found that pressure on reservoirs was rather somewhat higher than in the years before the pandemic [21,29,49]. Many factors contributed to this situation. One of the main concerns is the perception of angling as a safe activity, described above. The second is a lack of closure of open spaces by lockdown policies while closing many forms of indoor recreation as a result of implementing restrictions (less frequent closures relative to indoor spaces). Others are related to changes in amounts of leisure time because of lockdown policies. According to Mendiratta et al., job loss and changes in work characteristics (online work—reduced mobility and reduced working hours) in many situations resulted in increased leisure time resources although were often combined with less money to live on [49]. This may have increased fishing in general and the danger of an increased motivation for subsistence fishing. 

Stokes et al. and Mendiratta et al. noted that the increased importance of these needs increased during lockdown periods, especially among the groups with the lowest income statuses [29,49]. The increase in fishing activity among these demographic groups was also confirmed by our research. The anglers may have been forced to meet the subsistence needs caused by job loss, which was not present in the pre-pandemic period. However, this conclusion appears to be ambiguous. Midway et al. noted that angling for subsistence appeared to be of decreasing importance during the pandemic [21]. The key issue here may seem to be the average level of wealth of the surveyed group.

Midway et al. found that anglers with a small but statistically significant increase in fishing frequency caught more fish than before the pandemic. This would need to be looked at in terms of different causes [21]. More free time may also have resulted in longer stays on each of these trips. In particular, anglers who lived close to their fishing sites could effectively spend more time on the fishing activity itself. This is also a factor that potentially increases the level of pressure. Thus, increased leisure time and the identified need to escape from pandemic stress and media noise, which, according to Fullana et al., intensify pandemic fear, could be identified as factors that increase angler activity [60]. Simultaneously, it should be assumed that these factors will occur in different intensities in different social groups. In identifying them, it is worth noting the relationship of the proportion of each socio-demographic group of anglers more frequently angling in the pandemic relative to their proportion in the entire group of respondents—the percentage difference index (PDI). The more the magnitude of this index exceeds 1.0, the greater the importance of the group in potentially increasing pressure on fish stocks. 

In identifying the effect of more time on the increase in angler activity during the pandemic, one would need to consider the factor of angler age and work activity. For example, the increase in free time due to job loss may not have been experienced by the 65+ age category (6.4% of respondents in our study). Potentially, they would not have experienced the anxiety of job loss. In contrast, anxiety about an immediate life-threatening illness had the potential be the most stressful, which translated into the highest acceptance of vaccination. However, this group did not seek more frequent angling as they accounted for only 2.0% of the more frequent anglers in the pandemic (PDI 0.31). Based on data indicating that the populations of the “Western world” are aging and their economic statuses are deteriorating, as well as data indicating the aging of the angler population [61,62], the sense of security of older anglers should be considered crucial for the future development of angling.

Our respondent group was dominated by anglers who fished at least once a week or more often, with a total of 55.4%. In contrast, among those who reported more frequent fishing in the pandemic, the proportion of these anglers was over 75.5%. The ratio of these two quantities (PDI 1.36) indicates that the higher frequency of fishing was most characteristic of those who were already most intensively involved in fishing before the pandemic. Thus, it is reasonable to conclude that they already had a relatively large amount of leisure time and had the most strongly developed need to fish. Confirming the trend of increased angler fishing activity compared to the pre-pandemic season are the results of a study by Skrzypczak and Karpiński, who found similar fishing intensities among European anglers in 39% of respondents [37].

Among those fishing more frequently during the pandemic, there was an increase in the percentage of rural residents (PDI 1.17) and those in metropolitan areas with more than 100,000 residents (PDI 1.11). In the group of more frequent anglers, the rate of increase in the percentage share was slightly higher among anglers whose most frequented fishing spot was located in a city (PDI 1.09). It is worth noting that nearly 43% of such anglers lived in cities of more than 100,000 inhabitants. More than 71% of the anglers who fished more intensively during the pandemic and visited urban fishing grounds most commonly used fishing grounds located within 10 km of their place of residence. Among all those fishing more frequently in the pandemic, anglers commuting about 6–10 km to fishing grounds had the highest percentage difference index (PDI 1.47). At the same time, more than 61% of the more frequent fishers belonged to the socioeconomic group reporting angling expenditures in the range of EUR 100 to 500 (PDI 1.08). However, anglers spending between EUR 500 and 1000 had the highest percentage difference index (PDI 1.44). Among the more frequent fishing respondents, males aged 26–40 years predominated (44.9%), but they also dominated the study population (42.6%). The greatest increase in participation in more frequent angling compared to that of the entire group of respondents was observed in the youngest age category, i.e., those under 25 years old (PDI 2.12). The correlation of the anglers’ ages with their fishing experience confirms this pattern. More frequent fishing was more characteristic of less experienced anglers. For anglers with less than 5 years of angling experience, the ratio was 2.04. In the other angling experience categories, i.e., anglers with 5–10 years and 11–20 years of angling experience, the ratios were much lower at 1.20 and 1.16, respectively. Thus, it should be assumed that if more free time and the potential fear of losing one’s job are factors in the increase of angling activity, they concern mainly young people, i.e., those still at school and at the forefront of their professional career. According to Pinder et al., anglers who are young or have been fishing for a short time have the most detrimental impact on fish stocks [63]. Thus, the potential increase in the negative impacts of angling on fish stocks could be a very real consequence of the pandemic.

## 5. Conclusions and Managerial Implications

The prominence of angling in the perceived reduction of pandemic stress in anglers’ self-perception is undeniable. In contrast, the balance of benefits from changes in anglers’ pandemic behavior is ambiguous. This is determined by the potential increase in pressure on the environmental resources that anglers use. Nevertheless, the dynamics of anglers’ behavior and their responses in the social domain are very different.

Among the anglers who experienced reduced pandemic stress while fishing, the respondents with the least involvement in fishing (less spending, a lower frequency of fishing, and the least experience) were the anglers most probably concerned about SARS-CoV-2 infections and limited contact with friends and family. Meanwhile, anglers who did not report that fishing reduced their stress went fishing less often. However, less frequent fishing does not mean less time spent fishing. As mentioned above, anglers revealed changes in behavior during the pandemic by fishing closer to home. This could mean that they were averaging out fishing longer with the same or decreasing the number of trips. However, this is only a conjecture that should be clarified in future research.

Policymakers must weigh the disadvantages of restrictions against safety gained for society’s health at the national and local levels. Careful choices need to be made about how future lockdowns will affect society, the economy, and the environment because it seems obvious that the inherent rights of people to connect with nature cannot fall victim to the virus or other illnesses. Considering the established positive effects of angling on stress reduction, policymakers should not impose restrictions on access to fisheries, especially with best practices for reducing the impact of “sanitary bottlenecks” to reduce pandemic risks. Our research suggests the opposite—that there is no legitimate need for far-reaching restrictions on recreational fishing, either for economic or social reasons. However, the environmental impact of so-called pandemic angling is currently uncertain and should be an area of particular concern for researchers in the near future.

Our final conclusions are somewhat ambivalent. From the one side, it is important to remember that a low risk is a risk, and during a pandemic, all people should be aware of that. SARS-CoV-2 can be dangerous to everyone, even those vaccinated or recovered, and there are not many risk-free activities, even outdoors and in remote areas [31]. However, from the other side, knowing that recreational fishing is a multi-billion Euro industry [64], we can only repeat Ludwig von Mises, who said that “the source of all wealth and prosperity is production—not restriction. Any restriction should be decided upon after a detailed evaluation of its costs and benefits. No reasonable person can dispute this rule.” Furthermore, we can paraphrase him that there are, of course, moments when people may justify restrictions. For example, fire prevention regulations are restrictive and raise costs, but their price is justified by avoiding a larger disaster [65]. One is left to wonder: should we be afraid of such a fire during fishing?

## 6. Study Limitations

In examining anglers’ attitudes toward the pandemic and its limitations, the authors based their analysis on objective arguments that angling recreation is relatively safe, especially in the context of many other activities. At the same time, it should be noted that anglers are inherently positive about their hobby. Thus, they may be suggestive and subjectively oriented toward any aspect of their activity, regardless of the extent to which it reflects pandemic reality. There is a lack of information about susceptibility to anxiety and the causes of this fear. Vulnerability to pandemic stress may be due to many social, economic, and health factors that we did not ask about. The stressful impact of a pandemic may be due to fears of job and income losses (this will affect the economically active younger portion of the population). It may also be worsened by economic status (e.g., access to savings that can be drawn upon at a difficult time) and responsibility for those in the respondents’ care. An additional burden factor during a pandemic may be the presence of civilization diseases that contribute to increased mortality among those infected with SARS-CoV-2. Furthermore, we do not know to what extent the accumulation of COVID-19 stress factors may influence depressive states and decisions to abandon recreational activities among anglers. It should be taken into account that even if angling in the majority of respondents (63.8%) induces feelings of stress reduction, it does not exclude that for some anglers, stress levels translate into the abandonment of recreational activities. For this reason, among others, a separate part of the analysis focused on only the group of respondents who explicitly identified with a sense of pandemic stress reduction. It should be noted that our study was dominated by affiliated anglers (81.4%) but did not include an analysis of the potential relationships between and impacts between the affiliated group and environmental consequences and stress reduction. It might also be interesting in the future to explore non-anglers’ perspectives on this activity in terms of the institutional response to the pandemic crisis (e.g., do non-anglers consider this activity safe or appropriate in a pandemic, or have they changed their approaches to angling during the pandemic?).

## Figures and Tables

**Figure 1 ijerph-19-04346-f001:**
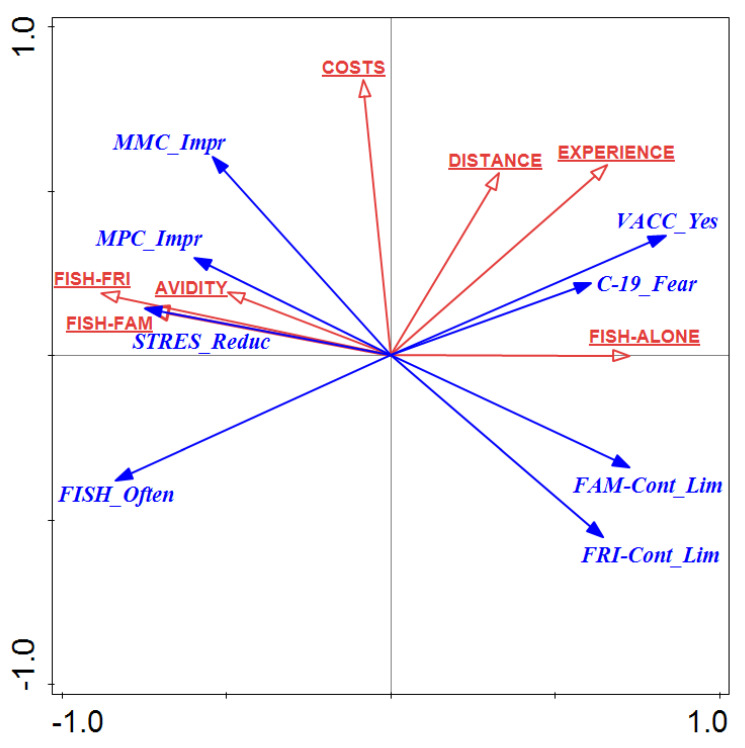
Ordination triplot of redundancy analysis (RDA) of anglers’ feelings and behavior induced by the COVID-19 pandemic (response data-blue arrows) in relation to their angling preferences and engagement (explanatory variables—red arrows. Both canonical axes are linear combinations of explanatory variables. Vectors pointing in the same direction indicate a positive correlation, vectors crossing at right angles indicate a near-zero correlation, while vectors pointing in opposite directions show a high negative correlation. The closer the group is to the center of the ordinal space, the more averaged the explanatory variables are. Abbreviations: STRES_Reduc, feeling of stress reduction by angling; MPC_Impr, feeling of improvement in physical condition; MMC_Impr, feeling of improvement in mental condition; VACC_Yes, positive attitude toward vaccination against COVID-19; C-19_Fear, feeling afraid of getting sick from COVID-19; FAM-Cont_Lim, limiting contact with family; FRI-Cont_Lim, limiting contact with friends; FISH_Often, fishing more often during the pandemic.

**Figure 2 ijerph-19-04346-f002:**
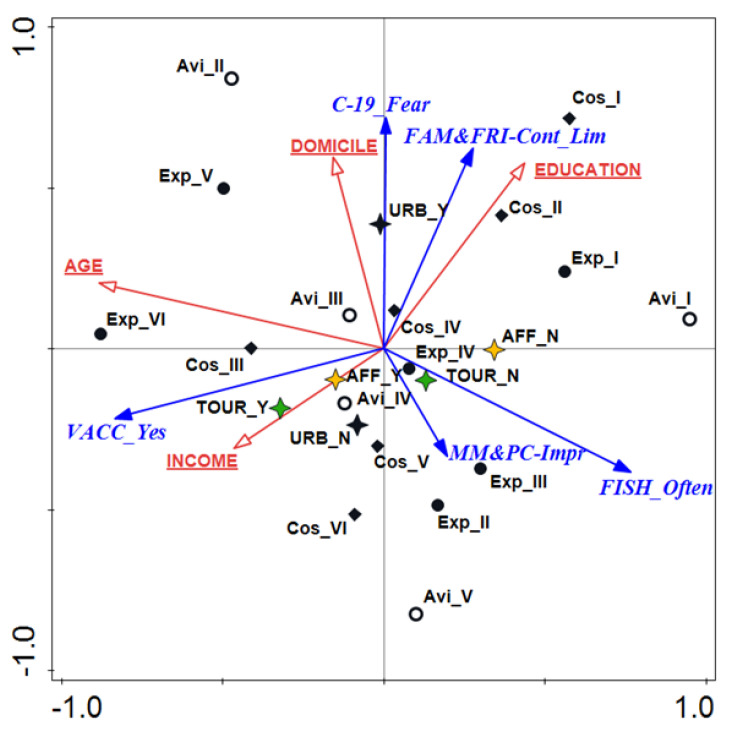
Triplot ordinal redundancy analysis (RDA) of reactions and behavior among anglers feeling positive effects of fishing on COVID-19 pandemic stress reduction (response data-blue arrows) versus socio-demographic factors (explanatory variables-red arrows). Abbreviations: VACC_Yes, positive attitude toward vaccination against COVID-19; C-19_Fear, feeling afraid of getting sick from COVID-19; FAM&FRI-Cont_Lim, limiting social contact with family and friends; FISH_Often, fishing more often during the pandemic; MM&PC_Impr, feeling of improvement in mental and physical conditions. For the anglers’ groups of preference and engagement abbreviations, see Appendix A (black circle-Experience categories; white circle-Avidity categories; black diamond-Cost categories; green star—two categories based on responses to the question “Fishing is the main tourism motive”, including positive-TOUR_Y and negative-TOUR_N; yellow star-two categories based on responses to the question “Affiliation in angling association”, including positive-AFF_Y and negative-AFF_N; black star-two categories based on responses to the question “Most visited angling spot is in the urban area”, including positive-URB_Y and negative-URB_N).

**Table 1 ijerph-19-04346-t001:** Perceptions and behavior toward angling and life attitudes during the pandemic period.

Characteristic	Response Rate on the Likert Scale * (in Percent)					Mean Score (±SD)
	1	2	3	4	5	
1. Fishing reduces my stress about the pandemic (STRES_Reduc)	11.7	4.8	19.7	22.3	41.5	3.77 ± 1.34
2. During the pandemic period, I fished more often (FISH_Often)	23.9	12.2	37.8	10.7	15.4	2.81 ± 1.33
3. I am concerned about getting sick (or getting sick again) from COVID-19 (C-19_Fear)	53.2	13.3	17.0	5.9	10.6	2.07 ± 1.38
4. I have a positive attitude towards vaccination against COVID-19 (VACC_Yes)	22.3	4.8	13.8	12.8	46.3	3.56 ± 1.61
5. My mental condition during the pandemic period has improved (MMC_Impr)	33.0	22.3	37.2	4.8	2.7	2.22 ± 1.04
6. My physical condition during the pandemic period has improved (MPC_Impr)	35.6	27.1	23.4	7.4	6.5	2.22 ± 1.19
7. I fish with my family (FISH-FAM)	25.0	22.3	25.0	12.2	15.4	2.71 ± 1.37
8. I have limited contact with my family during the pandemic period (FAM-Cont_Lim)	27.6	11.7	20.2	22.9	17.6	2.91 ± 1.46
9. I fish with my friends (FISH-FRI)	13.8	11.7	26.6	18.1	29.8	3.38 ± 1.38
10. I have limited contact with my friends during the pandemic period (FRI-Cont_Lim)	25.0	9.1	18.1	22.3	25.5	3.14 ± 1.52
11. I fish alone (FISH-ALONE)	12.8	13.8	32.4	23.4	17.6	3.19 ± 1.25
12. I feel exposed to COVID-19 infection at the fishing spot (C-19_Exp)	89.9	8.0	0.5	0.5	1.1	1.15 ± 0.55
13. I have limited outdoor activities during the pandemic period (OUT-Activ_Lim)	46.8	15.4	20.7	10.1	6.9	2.14 ± 1.30

* Typical five-point Likert scale, where “1” was the equivalent of “I strongly disagree”, “5” meant “strongly agree”, and “3” meant “I have no opinion, or it is difficult to determine” (neutral opinion).

**Table 2 ijerph-19-04346-t002:** Summary statistics for RDA of COVID-19 pandemic-induced feelings and behavior among socio-demographic groups of anglers (response data) versus their preferences and engagement in angling (explanatory variables, *VIF* * *<* 10).

Axes	1	2	3	4	Total Variance
Eigenvalues	0.5927	0.1396	0.0448	0.0216	1.000
Pseudo-canonical correlation	0.9516	0.9070	0.8157	0.8933	
Cumulative percentage variance					
of response data	59.27	73.23	77.71	79.87	
of fitted response data	72.97	90.16	95.68	98.33	
Sum of all eigenvalues					1.0000
Sum of all canonical eigenvalues					0.8122

* Variance inflation factor.

**Table 3 ijerph-19-04346-t003:** Differences between anglers’ groups with different stress reduction perceptions: with a noticeable reduction of pandemic stress through angling (Likert score answers 4–5; N = 360; MoE = 5.0%) and without it (Likert score answers 1–2; N = 93; MoE = 7.5%). Results are presented using the mean score ±SD.

Anglers’ Characteristics and Behavior	Feeling of Stress Reduction through Angling	
	Positive	No Effect
^1^ During the pandemic period, I fished more often	3.02 ± 1.36 ^A^	2.46 ± 1.21 ^B^
I am concerned about getting sick (or getting sick again) from COVID-19	2.12 ± 1.44	2.00 ± 1.26
^2^ I have a positive attitude towards vaccination against COVID-19	3.48 ± 1.66 ^A^	3.71 ± 1.53 ^B^
My mental condition during the pandemic period has improved*	2.10 ± 1.06	2.29 ± 1.06
My physical condition during the pandemic period has improved*	2.14 ± 1.22	2.39 ± 1.32
^3^ I have limited contact with my family during the pandemic period*	3.00 ± 1.55 ^A^	2.55 ± 1.33 ^B^
I have limited contact with my friends during the pandemic period*	3.14 ± 1.65	3.03 ± 1.50
I fish with my family	2.74 ± 1.39	2.71 ± 1.23
I fish with my friends	3.48 ± 1.42	3.45 ± 1.41
I fish alone	3.23 ± 1.27	3.12 ± 1.20
I feel exposed to COVID-19 infection at the fishing spot	1.13 ± 0.52	1.18 ± 0.59
^4^ I have limited outdoor activities during the pandemic period	2.25 ± 1.38 ^A^	1.81 ± 1.19 ^B^

Values with various superscripts (^A^;^B^) are significantly different using the non-parametric Kruskal–Wallis test (*N = 453, df* = 1); *^1^ H = 4.17, p = 0.0411; ^2^ H = 3.91, p = 0.0479; ^3^ H = 4.65, p = 0.0353; ^4^ H = 5.61, p = 0.0178.*

**Table 4 ijerph-19-04346-t004:** Summary statistics for RDA of anglers’ reactions and behavior and perceived positive effects of angling on COVID-19 pandemic stress reduction (response data) versus selected socio-demographic factors (explanatory variables, *VIF* * *<* 10).

Axes	1	2	3	4	Total Variance
Eigenvalues	0.7687	0.0569	0.0093	0.0755	1.000
Pseudo-canonical correlation	0.9585	0.7458	0.3903	0.0000	
Cumulative percentage variance					
of response data	76.87	82.56	83.49	91.04	
of fitted response data	92.07	98.89	100.00		
Sum of all eigenvalues					1.0000
Sum of all canonical eigenvalues					0.8349

* Variance inflation factor.

## Data Availability

The data that support the findings of this study are available upon request from the corresponding author, and are in the Polish language.

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
