# Peer review of "The Significance of Angling in Stress Reduction during the COVID-19 Pandemic—Environmental and Socio-Economic Implications"

_ijerph, 2022, doi:10.3390/ijerph19074346_

Round 1

Reviewer 1 Report

I only completed partial review, but it needs improvement before publishing. See my edits.

Author Response

Thank you for your review. Please see the attached file for our response to Your comments and suggestions.

Reviewer 2 Report

Thank you for the opportunity to review this manuscript.

  I have identified some areas to help improve the manuscript and make it even better:  
  • recommend a grammar review for clarity/flow of sentences.
  • lines 39, 'regime' may want to also include 'public health' to be more applicable to readers in multiple countries.
  • recommend making lines 292 - 394 into multiple paragraphs for clarity and readability.
  • please change section 6 to plural (study limitations).
  • areas for future research could also be listed that include:
    • surveying non-anglers to also identify their perceptions on the activity as related to pandemic public health initiatives.
    • identifying years of experience as a variable for surveyed anglers.
  • Please ensure all COVID acronyms are in all caps.
  • in the Introduction, additional comments surrounding the actual pandemic-related threats of exposure for anglers can be expanded. For instance, what pandemic exposure risks are out there that did not exist pre-pandemic? What's the difference(s) between these two time periods?
  This article and the statistical analyses are very well done. I am so glad to see a study on fishing and mental health/etc. Wonderful manuscript. Please entertain these suggestions and very good job.

Author Response

(The authors gave the same response as above.)

Reviewer 3 Report

Manuscript No.: ijerph-1644119

Manuscript Title: The significance of angling in stress reduction during the COVID-19 pandemic - environmental and socio-economic implications

The quality of the paper is good. The English language is good and the methodology section is clear. However, the authors should keep in mind that the paper should be understandable for people who didn’t participate in this study and unfortunately that is still not the case. For example, the fourth paragraph of introduction section mentions "relatively safe" suddenly. But it is unclear to me why you sudden bring the "relatively safe "before talking about the increased interest in fishing in the last paragraph. And there are more unclarities in the paper.

Second, only two or three sentences cannot be a paragraph. There are so many paragraphs like this. Please revise or combine them accordingly.

Third, furthermore, I still have difficulties with the scientific basis of this research and the logical structure of this manuscript. The emphasized part of theoretical foundation is a bit strange here, for example. What is the connection between the theoretical foundation and literature review sections?

Fourth, the literature review part is missing. Readers cannot see the previous related studies on the topic of environmental and socio-economic implications of COVID19.

Finally, I don’t understand how you come to the items of Table 3. I don’t think it is professional for researchers. At minimum, one specific characteristic defines a good list of response options for survey questions. For example, the categories (response options) must be mutually exclusive, which means they do not overlap with one another. Page 7, Table 3, you can see while several response lists are exhaustive, it does not provide mutually exclusive categories. My mental and physical condition during the pandemic period has im-proved, etc., for example. If I only care about my mental health or condition, how can I choose from your items? The authors are highly encouraged to completely recheck the information in the description of the research design, redo the data analysis, present the results in tables, etc.

Author Response

(The authors gave the same response as above.)

Round 2

Reviewer 3 Report

Manuscript No.: ijerph-1644119

Manuscript Title: The significance of angling in stress reduction during the COVID-19 pandemic - environmental and socio-economic implications

I am pleased to see that the manuscript has improved significantly since the first time I reviewed it. However, there remain several outstanding issues that I believe are necessary to address before publication. 

There are numerous typographical and grammatical errors throughout the text that must be corrected before the paper is accepted. 

I strongly recommend that the authors use a different Figure to display the ordination triplot of redundancy analysis. Not only is the Figure very ugly to look at, it also makes the data difficult to read / understand. 

Again, I also strongly recommend that authors could add the section of literature review.

Once these issues are address, I will be happy to recommend this excellent study for publication. 

Author Response

Thank you very much for your comprehensive review. Please see the file attached
